# OpenReview forum: "To Distill or Not to Distill: Knowledge Transfer Undermines Safety of LLMs"
_ICLR.cc/2026/Conference — ICLR 2026 Conference Withdrawn Submission_

### Official Review · Reviewer_LVSz · 2025-10-29

**Soundness:** 2
**Presentation:** 3
**Contribution:** 2
**Rating:** 2
**Confidence:** 4

**Summary:**

This paper conducts experiments to show that safety on small models will be undermined with model distillation. It is an intuitive guess, and the authors have verified on both soft and hard labels. Uncertainty and semantic drift have been analyzed in distillation. Three models have been used to justify safety in three tasks: machine translation, arithmetic reasoning, and medical instruction following. Code has been released.

**Strengths:**

1. It is an intuitive viewpoint that safety will be undermined with model distillation. All types of performance should have been undermined. That is why it is called distillation.
2. The reviewer appreciate the authors' effort to conduct extensive experiments.
3. The analysis could benefit future technology of model distillation.

**Weaknesses:**

1. This paper is more of an empirical study paper rather than a research paper proposing new methods. While extensive experiments have been conducted, no new method has been proposed. With these analyses, the reviewer would expect new distillation methods to address the issue. The current state of the paper may fit other tracks, such as position paper tracks or the benchmark track.

2. The impact of this paper could be limited. When fine-tuning with human feedback is dominating in both industry and academia to address safety, safety with model distillation would not excite readers, as it is the inherent weakness of model distillation. Even after model distillation, fine-tuning is still necessary, which may hinder the contribution of the observation.

3. The paper introduces tasks including machine translation, arithmetic reasoning, and medical instruction following. From the perspective of the reviewer, the safety in machine translation and arithmetic reasoning is not the biggest concern. The paper should introduce more safety-related benchmarks, such as [1].

4. Presentation could be improved. For example, Figure 6 and Table 2 are out of the paper border, which could violate the format requirement.

[1] SALAD-Bench: A Hierarchical and Comprehensive Safety Benchmark for Large Language Models

**Questions:**

NA

---

> ### Author Response · Authors · 2025-12-03
>
> Thank you for taking the time to read and provide feedback on our paper. We will address your concerns in order.
>
> > **S1:** It is an intuitive viewpoint that safety will be undermined with model distillation. All types of performance should have been undermined. That is why it is called distillation.
> >
>
> While you highlight this as a strength of the paper, it is actually not intuitive that safety will be undermined with model distillation. In fact, learning the probability distributions of larger, safer language models would intuitively suggest safer resultant models. Similarly, your claim that "all types of performance should have been undermined. That is why it is called distillation" is unclear and certainly not the case with knowledge distillation using soft labels. To the contrary, distillation generally creates more specialized and robust models.
>
> > **W1:** This paper is more of an empirical study paper rather than a research paper proposing new methods. While extensive experiments have been conducted, no new method has been proposed. With these analyses, the reviewer would expect new distillation methods to address the issue. The current state of the paper may fit other tracks, such as position paper tracks or the benchmark track.
> >
>
> We would like to **strongly dispute** the claim that an empirical study is not a research paper. Our work does not release a new algorithm or method, but it does contribute to the field with its novel analysis of a highly relevant and widely used training paradigm (knowledge distillation). We aim for these analyses to lead to better informed training methods and decisions.
>
> Regarding track selection: a) authors do not pick the track, rather the domain (AI Safety and Alignment in our case), and b) position papers argue a specific viewpoint on a relevant topic, supported by reasoning and evidence from previous literature, to stimulate discussion rather than present new experimental results. In our case, we have spent hundreds of GPU hours and FLOPS to distill and evaluate the models.
>
> > **W2:** The impact of this paper could be limited. When fine-tuning with human feedback is dominating in both industry and academia to address safety, safety with model distillation would not excite readers, as it is the inherent weakness of model distillation. Even after model distillation, fine-tuning is still necessary, which may hinder the contribution of the observation.
> >
>
> The assumption that all instruction-tuned models are safe and deployment-ready (as often mentioned in their technical reports) encourages practitioners to use models out-of-the-box. While applying fine-tuning with benign/harmless data has shown slight reductions in safety (inspiring new methods and awareness in the community), the same scrutiny has not been applied to knowledge distillation. This paper sheds light on that matter, now encouraging safety-aware distillation or post-distillation safety training on instruction-tuned models.
>
> > **W3:** The paper introduces tasks including machine translation, arithmetic reasoning, and medical instruction following. The paper should introduce more safety-related benchmarks, such as [1].
> >
>
> This is **not a benchmark paper**, hence we do not introduce any new datasets. We do not introduce the mentioned tasks; rather, we use existing datasets to conduct knowledge distillation for different types of utility. Similarly, we use pre-eminently established safety benchmarks across three dimensions: toxicity (RealToxicityPrompts), faithfulness/hallucinations (FaithEval), and jailbreak (JBBench) as tools to identify shifts in safety after knowledge distillation. To reiterate, our work is a post-hoc empirical study examining the safety-utility trade-off within knowledge distillation and identifying possible causes.
>
> > **W4:** From the perspective of the reviewer, the safety in machine translation and arithmetic reasoning is not the biggest concern.
> >
>
> All state-of-the-art distillation methods introduced as iterative improvements and recently published at ICML, ICLR, and NeurIPS (MiniLLM, DistilLLM, GKD, DistillLM-2, Speculative KD, etc.) use similar datasets (refer to their papers, where machine translation and arithmetic reasoning are consistently present) to showcase performance improvements. The assumption made and understood in the community is that these improvements generalize beyond to other domains. Moreover, along with machine translation and arithmetic reasoning, we also use medical instruction following, a highly critical and important use case as a training dataset.

---

### Official Review · Reviewer_NpKh · 2025-11-02

**Soundness:** 2
**Presentation:** 3
**Contribution:** 2
**Rating:** 4
**Confidence:** 4

**Summary:**

This paper studies an under-examined but important question: whether knowledge distillation (KD) erodes the safety of LLMs more than supervised fine-tuning (SFT). The authors compare five post-training regimes: SFT, sequence-level KD (hard labels), and three soft-label / KL-based distillation methods (vanilla KD with forward KL, MiniLLM with reverse KL, and on-policy GKD with JS divergence) - across three benign tasks (Marathi to English translation, MetaMathQA arithmetic reasoning, and MedInstruct medical IF). They then evaluate all resulting students on three safety dimensions: jailbreak robustness (JailbreakBench+PAIR), toxicity (RealToxicityPrompts), and faithfulness/hallucinations (FaithEval). The central empirical finding is sharp: soft-label, logits-based distillation gives the biggest utility gains but degrades safety the most—up to ~50 percentage points more than SFT on the same benign data, and far more than hard-label SFT. The paper further offers a mechanistic story: logit-based KD causes asymmetric uncertainty shifts—students stay confident on the trained utility tasks but become more epistemically uncertain specifically on safety queries, which correlates with more jailbreak successes. A semantic-drift analysis with unbalanced OT shows that being closer to the teacher semantically can paradoxically correlate with worse jailbreak outcomes, suggesting KD may also transfer undesirable lexical/safety patterns. The paper argues that, as KD becomes standard in open LLM pipelines, we need safety-aware distillation objectives.

**Strengths:**

1. Clear problem formulation. KD is widely used in open-source LLM training, but most safety work has focused on SFT/RLHF; showing that “KD is not safety-free” is a useful corrective.

2. Systematic comparison of multiple KD flavours. Evaluating SFT, SeqKD, FKLD (KD), RKLD (MiniLLM), and JS/on-policy GKD on the same benign tasks and multiple model families (Qwen2, Llama-3.2, Gemma3) makes the conclusion harder to dismiss as “it was just one setup.” Table 3 is especially informative in this regard.

3. Multidimensional safety evaluation. Many papers stop at jailbreaks; this one also measures toxicity and faithfulness, and finds the pattern is fairly consistent—distillation hurts more often than not.

4. Interesting mechanism section. The logit-evidence / Dirichlet-based uncertainty analysis (LogTokU) to explain why safety collapses after KD is novel in this context and helps connect the empirics to a plausible explanation (safety tokens become high-evidence but low-knowledge → brittle refusal).

5. Practical takeaway. The result that pretrained (non-IT) students preserve safety better than IT students after KD is useful guidance for practitioners who want to compress guard-like models.

**Weaknesses:**

1. Limited contribution. Many prior work has already studied and discovered that KD can erode safety, or more broadly trustworthiness. For example, [1][2] have surveyed many work on privacy, safety, etc. There is no significant difference of this work.

2. No baselines for safety-aware KD methods. Some KD methods are safety-aware. This paper only include non-safety-aware KD methods.  Please see [3][4]. The paper does not try the simplest ones (KD + refusal loss; KD + safety LoRA frozen; distill logits only on utility tokens but cross-entropy on safety tokens). Even a small experiment here would strengthen the contribution.

3. Causality of "soft labels" leads to "safety loss" is not fully pinned down. The paper observes that KL-based KD correlates with stronger safety degradation, but several plausible co-factors are entangled: (i) teacher-student shared tokenizer and lineage; (ii) offline vs on-policy setting; (iii) specific benign tasks (translation is repeatedly noted to be “catastrophic” for jailbreaks); and (iv) warm-up SFT before KD for some methods. It is therefore hard to say “KD causes it,” as opposed to “this very common recipe for KD causes it.”

4. Limited task diversity on the training side. All training data are “benign” and quite structured (translation, math, medical IF). When we distill LLMs, we usually include dialogue or long-form instruction data with mixed safety content. It is possible that KD on mixed-safety or safety-aware corpora would look less negative; the paper only hints at this (Sec. 5.1-5.3) but does not test it.

5. Safety metrics are all output-side. The paper argues in Sec. 5.2 about "lexical pattern reinforcement" as a mechanism, but the presented evidence is indirect (semantic UOT correlations and uncertainty shifts). There is no representation-level or probing-level evidence (e.g., safety head activations, refusal-token margin analyses) to directly show that KD overwrites alignment features.

6. Interpretation of the UOT result is a bit speculative. The finding that "closer to teacher leads to more jailbreaks" is interesting but could also be an artefact of (a) teacher itself not being perfectly safe; (b) students over-fitting to teacher phrasing that jailbreak benchmarks exploit; or (c) distribution mismatch between evaluation and distillation outputs. A brief ablation where the teacher is more strongly aligned (e.g., a Safety-RLHF-trained teacher) would make this claim more solid.

References
[1] A Comprehensive Survey on Knowledge Distillation, Mansourian et al., TMLR, 2025
[2] A survey on knowledge distillation: Recent advancements, Moslemi et al., 2024
[3] DistillSeq: A Framework for Safety Alignment Testing in Large Language Models using Knowledge Distillation, Yang et al., arxiv, 2024
[4] Complementary KD for robust & privacy-preserving VFL, Gao et al., AAAI, 2024

**Questions:**

1. How teacher-dependent is the safety drop? If you distill from a more aligned but slightly less capable teacher (e.g. an RLHF’d version of the same model), does KD still degrade safety more than SFT, or does the effect shrink?

2. Is it possible to decouple "task effect" from "KD effect"? The authors mentioned that translation triggers the worst safety deterioration across all methods. If you SFT on translation but KD on arithmetic (or vice versa) for the same student, do you still see the same pattern? A small cross-task swap experiment would clarify whether some of your results are task-driven.

3. Why does semantic proximity to the teacher predict worse jailbreak resistance when the teacher is better? This is counter-intuitive and central to your argument in Sec. 4.2. Can you show at least one qualitative example where the student “copies” a teacher structure but loses the refusal token?

4. Did you try KD with temperature > 1 at training time specifically to flatten teacher distributions? "temperature is set to 1" for student training; but much of the brittleness you observe could be a consequence of over-confident targets.

5. How stable are the results across random seeds and judge models? JailbreakBench + PAIR can be sensitive to the refusal style. Did you verify with a second LLM judge or with a toxicity classifier different from Detoxify?

6. Could safety be restored cheaply post-KD? You discuss RESTA/LISA/SafeMERGE as heavy-weight options. What happens if we just do a small, 2-5k example safety SFT after KD-do we recover SFT-like safety while keeping (some of) KD utility? That would make the paper even more actionable.

---

> ### Author Response · Authors · 2025-12-03
> **Addressing Factual Inaccuracies in the Review**
>
> We thank you for taking the time to read our work and provide feedback. We address your concerns below in order.
>
> > Limited contribution. Many prior works have already studied and discovered that KD can erode safety, or more broadly trustworthiness. For example, [1][2] have surveyed many works on privacy, safety, etc. There is no significant difference of this work.
> >
>
> **We strongly object to this claim.** This assertion is **factually** **incorrect**. To our knowledge, **no previous study** has demonstrated that knowledge distillation erodes safety in large language models. We have carefully reviewed the papers you cite:
>
> - Paper [1] (Mansourian et al., TMLR 2025) is a general survey on knowledge distillation techniques. It does **not** discuss safety erosion, alignment degradation, or jailbreak vulnerabilities in the context of LLM distillation.
> - Paper [2] (Moslemi et al., 2024) similarly surveys distillation advancements but does **not** contain **any** analysis of safety degradation in distilled LLMs.
>
> Neither paper addresses the core question of our work: whether distillation causes safety degradation beyond what is observed in standard supervised fine-tuning on the same benign data. **This is the first systematic study** to compare multiple distillation objectives (forward KL, reverse KL, on-policy GKD) against SFT across multiple safety dimensions (jailbreaks, toxicity, faithfulness) and multiple model families, demonstrating that soft-label distillation incurs a substantial safety tax. The claim that there is "no significant difference" from prior work is unsupported and mischaracterizes both the cited literature and our contribution.
>
> > No baselines for safety-aware KD methods. Some KD methods are safety-aware. This paper only include non-safety-aware KD methods. Please see [3][4]. The paper does not try the simplest ones (KD + refusal loss; KD + safety LoRA frozen; distill logits only on utility tokens but cross-entropy on safety tokens). Even a small experiment here would strengthen the contribution.
> >
>
> We must respectfully disagree, as there is currently no literature focused on distillation loss functions that are safety-aware. Our findings aim to inspire new methods in this area.
>
> Regarding the papers you mention, we believe there may be a misunderstanding of their scope and their methods. Paper [3] focuses on distilling and creating a small model that can generate jailbreak attacks on larger LLMs more efficiently. Paper [4] presents a method specific to neural networks (DeepFM) for tabular data, whereas our work pertains to language models. Both unrelated to what you cite them for.
>
> Currently, no distillation training methods prioritise safety alongside utility. Our work aims to demonstrate the elevated levels of safety degradation observed due to distillation and identify possible causes, which we hope future work will address.

---

> ### Author Response · Authors · 2025-12-03
> **Response to Weaknesses**
>
> > Causality of "soft labels" leads to "safety loss" is not fully pinned down. The paper observes that KL-based KD correlates with stronger safety degradation, but several plausible co-factors are entangled: (i) teacher-student shared tokenizer and lineage; (ii) offline vs on-policy setting; (iii) specific benign tasks (translation is repeatedly noted to be "catastrophic" for jailbreaks); and (iv) warm-up SFT before KD for some methods.
> >
>
> We appreciate this methodological concern and address each co-factor:
>
> **(i) Shared tokenizer and lineage:** Our experimental design controls for this by comparing methods within the same model family (e.g., all Qwen2 experiments use identical teacher-student tokenizer pairs). The key comparison is between methods using the same teacher-student pair but different loss functions (SFT vs SeqKD vs KD/MiniLLM/GKD), which isolates the effect of soft vs hard labels.
>
> **(ii) Offline vs on-policy:** We explicitly evaluate both settings. Table 3 shows that offline methods (KD, MiniLLM) and on-policy method (GKD) exhibit similar safety degradation patterns, suggesting the offline/online distinction is not the primary driver. In fact, GKD often shows comparable or worse safety degradation despite being on-policy.
>
> **(iii) Task-specific effects:** We agree translation shows particularly severe degradation. However, the pattern of "soft-label distillation degrades safety more than SFT" holds consistently across all three tasks (translation, arithmetic reasoning, medical IF), as shown in Table 3. The magnitude varies by task, but the directional effect is consistent, indicating this is not merely a task artifact.
>
> **(iv) Warm-up SFT:** All distillation methods (SeqKD, KD, MiniLLM, GKD) receive identical one-epoch warm-up fine-tuning before three epochs of distillation, ensuring fair comparison. SFT trains for four epochs total on the same data. This standardized protocol isolates the distillation effect.
>
> Our ablations in Table 2 further support causality: the safety-utility ratios differ dramatically between hard-label (1.857 for SeqKD) and soft-label methods (0.171-0.275), despite identical training data and procedures except for the loss function.
>
> > Limited task diversity on the training side. All training data are "benign" and quite structured (translation, math, medical IF). When we distill LLMs, we usually include dialogue or long-form instruction data with mixed safety content.
> >
>
> This is a deliberate design choice, not a limitation. We specifically use purely benign data to provide a controlled setting for measuring safety degradation, following established methodologies. Including mixed-safety data would confound the analysis: any observed safety degradation could be attributed to explicit unsafe content rather than the distillation process itself.
>
> Our tasks span three distinct domains with different characteristics:
>
> - **Translation:** Structured, deterministic, cross-lingual
> - **Arithmetic reasoning:** Multi-step reasoning with verifiable answers
> - **Medical IF:** Open-ended, knowledge-intensive, safety-critical domain
>
> These tasks are representative of common distillation use cases in production systems (Llama 4, Gemma3, DeepSeek all use similar task distributions). Our finding that even benign data causes substantial safety degradation through distillation is more concerning than if it only occurred with mixed-safety data.
>
> The reviewer's suggestion that "KD on mixed-safety or safety-aware corpora would look less negative" is precisely the research direction we hope to inspire. Our work establishes the baseline problem; developing safety-aware distillation methods is valuable future work that our findings motivate.

---

> ### Author Response · Authors · 2025-12-03
> **Response to Weaknesses (2)**
>
> > **Safety metrics are all output-side.** The paper argues in Sec. 5.2 about "lexical pattern reinforcement" as a mechanism, but the presented evidence is indirect (semantic UOT correlations and uncertainty shifts). There is no representation-level or probing-level evidence (e.g., safety head activations, refusal-token margin analyses) to directly show that KD overwrites alignment features.
> >
>
> We acknowledge this is a valid point. Our mechanistic analysis combines multiple approaches:
>
> 1. **Token-level uncertainty (Section 4.1, Table 4):** We use LogTokU to analyze epistemic and aleatoric uncertainty at the token level, which provides insight into the model's internal confidence states. This shows soft-label methods produce 10× higher epistemic uncertainty on safety evaluations.
> 2. **Semantic drift analysis (Section 4.2, Figure 5, Table 11):** We use Unbalanced Optimal Transport to measure distributional changes in embedding space, revealing counter-intuitive patterns where semantic proximity to teachers correlates with worse safety.
>
> While we agree that representation-level probing (e.g., safety head activations, refusal-token margins) would strengthen our mechanistic story, such analyses require:
>
> - Identifying which attention heads constitute "safety heads" (no established methodology exists for distilled models)
> - Defining refusal-token sets that generalize across model families with different tokenizers
> - Computational resources for full activation analysis across 8 models and 3 tasks
>
> Our current analyses provide convergent evidence for the lexical pattern reinforcement hypothesis through: (a) output-level semantic shifts, (b) uncertainty patterns, and (c) correlation with safety outcomes. We acknowledge this limitation and suggest representation-level analysis as important future work. The output-level evidence, while indirect, is sufficient to establish the safety degradation phenomenon and propose plausible mechanisms.
>
> > **Interpretation of the UOT result is a bit speculative.** The finding that "closer to teacher leads to more jailbreaks" is interesting but could also be an artefact of (a) teacher itself not being perfectly safe; (b) students over-fitting to teacher phrasing that jailbreak benchmarks exploit; or (c) distribution mismatch between evaluation and distillation outputs.
> >
>
> We appreciate this methodological concern. Let us address each alternative explanation:
>
> **(a) Teacher safety:** Our teachers are instruction-tuned models from reputable sources (Qwen2-7B-Instruct, Llama-3.2-3B-Instruct, Gemma3-4B-Instruct) with established safety alignment. Table 3 shows teacher jailbreak success rates of 39.4-41.5%, which are substantially lower than the 50-95% observed in distilled students. If teacher imperfection were the primary driver, we would expect students to plateau at teacher safety levels, not substantially exceed them.
>
> **(b) Overfitting to exploitable phrasing:** This is a plausible mechanism that actually supports rather than contradicts our lexical pattern reinforcement hypothesis (Section 5.2). Jailbreak benchmarks like JailbreakBench+PAIR are specifically designed to test robustness against adversarial phrasing. If students overfit to teacher phrasing that these benchmarks exploit, this demonstrates precisely the unsafe pattern transfer we posit. The correlation between semantic proximity (lower UOT distance) and worse jailbreak outcomes (Table 11, r=0.576, p<0.001) provides evidence for this mechanism.
>
> **(c) Distribution mismatch:** Our UOT analysis specifically accounts for distributional differences by measuring transport costs in embedding space. The UOT framework with L2 divergence penalties (Section 2.6, Appendix F) is designed to handle emergence/disappearance of semantic patterns. Moreover, the statistical significance (p<0.001) across 96 model combinations suggests this is not a random artifact.
>
> The counter-intuitive nature of this finding (semantic proximity predicting worse safety) is precisely why we present it with appropriate statistical evidence and acknowledge alternative mechanisms. It suggests distillation may transfer not just knowledge but also subtle linguistic patterns that undermine safety alignment.

---

### Official Review · Reviewer_UdH2 · 2025-11-02

**Soundness:** 3
**Presentation:** 2
**Contribution:** 2
**Rating:** 4
**Confidence:** 3

**Summary:**

This paper studies the safety implications of knowledge distillation from larger LLMs to smaller ones. The main finding is that supervised fine-tuning (SFT) better preserves safety than distillation. Soft-label KD achieves a better safety–utility trade-off than hard-label KD, yet still shows a significant safety retention gap compared to SFT.

**Strengths:**

The paper presents an analysis across models, KD methods, tasks, and evaluation metrics.

Additional post-hoc analyses are provided, offering two main explanations (with evidence) for why KD degrades safety more than SFT.

The takeaway derived from the analysis is interesting.

**Weaknesses:**

The analysis is interesting but not surprising, given prior work showing that SFT can improve utility while reducing safety even on harmless data. This paper confirms that KD tends to improve utility even more but also suffers greater safety degradation. Since the stronger utility of KD is already well known, the main takeaway reduces to "KD incurs more safety degradation"

Given the above weaknesses, I would expect at least a preliminary solution. As the first paper focusing specifically on this problem, even a simple proof-of-concept approach, ideally grounded in the analysis in Sec. 4, would make the contribution more actionable and inspiring for follow-up work.

In Fig. 2, while the evaluation pipeline is clear, the 2nd and 4th blocks contain redundant information.

Fig. 3 is hard to follow; the KD types are not consistently shown across all models.

All figures and tables: Provide more experimental-setting details and add a concise descriptive sentence to guide the reader.

**Questions:**

See weakness

---

> ### Author Response · Authors · 2025-12-03
>
> Thank you for taking the time to review our work. Following your suggestions, we have updated the figures and tables, along with their captions, to provide clearer and more descriptive guidance for readers.
>
> **Regarding preliminary solutions:**
>
> > Given the above weaknesses, I would expect at least a preliminary solution.  As the first paper focusing specifically on this problem, even a simple proof-of-concept approach, ideally grounded in the analysis in Sec. 4, would make the contribution more actionable and inspiring for follow-up work.
> >
>
> We appreciate this feedback. As the first work exploring the safety consequences of benign knowledge distillation, we have approached similar objectives to those considered in previous studies on safety-utility trade-offs in fine-tuning [1-4]. This includes identifying the degree of trade-off and exploring possible explanations through semantic shifts and increased uncertainty.
>
> Given the extent of our model training (8 models across 3 tasks), our scope was necessarily focused on analysing current state-of-the-art distillation techniques. We hope that our findings and post-hoc analysis will provide a foundation for developing methods to address the issues we have highlighted.
>
> [1] Qi, X., Zeng, Y., Xie, T., Chen, P. Y., Jia, R., Mittal, P., & Henderson, P. (2023). Fine-tuning aligned language models compromises safety, even when users do not intend to!. arXiv preprint arXiv:2310.03693.
>
> [2] He, L., Xia, M., & Henderson, P. (2024). What is in your safe data? identifying benign data that breaks safety. arXiv preprint arXiv:2404.01099.
>
> [3] Guan, Z., Hu, M., Zhu, R., Li, S., & Vullikanti, A. (2025). Benign samples matter! fine-tuning on outlier benign samples severely breaks safety. arXiv preprint arXiv:2505.06843.
>
> [4] Fraser, K. C., Dawkins, H., Nejadgholi, I., & Kiritchenko, S. (2025). Fine-Tuning Lowers Safety and Disrupts Evaluation Consistency. arXiv preprint arXiv:2506.17209.

---

### Official Review · Reviewer_sdkg · 2025-11-04

**Soundness:** 1
**Presentation:** 2
**Contribution:** 2
**Rating:** 2
**Confidence:** 3

**Summary:**

The paper argues that fine-tuning language models using knowledge distillation makes them compromise their safety performance more than fine-tuning them on standard fine-tuning datasets For knowledge distillation the authors utilize different experimental setups: hard label distillation, soft label distillation with forward KL divergence, reverse KL divergence, and JS divergence. Building on this observation, the authors argue that practitioners should be careful while using knowledge distillation for fine-tuning smaller models.

**Strengths:**

The authors compare multiple baselines of different ways to perform knowledge distillation. They also consider multiple tasks and open source their code for reproducibility.

**Weaknesses:**

It is not clear why distillation should lead to larger decrease in model’s safety performance as compared to standard fine-tuning. It seems that the key underlying reasons might be related to the distribution shift between the distillation and the SFT datasets. For some reason the SFT datasets seem to be difficult to train with (as the utility doesn't increase much on in-distribution datasets in table-3). It is also not clear why authors observe degradations in utility as well as safety performance (in some cases) in Table-3 on training using SFT. This might be because learning the SFT distribution is difficult for the model. Overall, I think the authors should look at the utility vs safety trade-off which they observe (in Fig.1 and Table-3) instead of highlighting the larger degradation in safety performance on using knowledge distillation. A fair comparison should have similar utility.

It would be great if the authors can bring some more insights on why they think knowledge distillation could hamper safety performance more than SFT. Currently, it is not clear as in case of using hard labels for knowledge distillation (KD), both SFT and KD would use the same training loss. I think deeper investigation is needed in this direction to make the claims of this paper more grounded.

**Questions:**

It would be great if the authors can try to use hard label distillation between different families of models. This can help them give effects similar to SFT.

I would recommend that the authors should try to compare their numbers for the same utility performance. Then their results can give a stronger signal.

Perhaps reframing their results as demonstrating safety-utility trade-off could be more interesting. This has also been observed recently in a few other works [1]

[1] The Jailbreak Tax: How Useful are Your Jailbreak Outputs?  (https://arxiv.org/abs/2504.10694)

---

> ### Author Response · Authors · 2025-12-03
> **Clarification on Safety-utility Tradeoff Narrative**
>
> We would like to thank you for taking your valuable time to review our work. Your feedback has been taken into consideration, and we will address the concerns and questions raised by you in a sequential manner.
>
> > It is not clear why distillation should lead to larger decrease in model's safety performance as compared to standard fine-tuning. It seems that the key underlying reasons might be related to the distribution shift between the distillation and the SFT datasets. For some reason the SFT datasets seem to be difficult to train with (as the utility doesn't increase much on in-distribution datasets in table-3).
> >
>
> Distillation leads to a larger decrease in models' safety performance as a consequence of additional information gain brought about by the soft probabilities, which result in higher utility gain at a higher safety cost (trade-off).
>
> A clarification on the datasets and distillation methods: SFT uses ground-truth labels provided with the dataset. SeqKD, the method that doesn't see in-distribution performance increase, uses teacher-generated outputs (labels) during distillation. Hence, we observe that while hard label distillation (SeqKD) and soft label distillation (KD, MiniLLM, GKD) all use the same dataset (and distribution), they result in varying levels of safety and utility shifts. Rather than a distribution-induced effect, we see that using cross-entropy with top-1 logit (SeqKD, SFT), instead of the entire vocabulary with KLD loss functions, results in poor safety and utility gains, while using the entire vocabulary with the same labels results in an increase in utility with greater safety loss.
>
> > It is also not clear why authors observe degradations in utility as well as safety performance (in some cases) in Table-3 on training using SFT. This might be because learning the SFT distribution is difficult for the model.
> >
>
> As mentioned above, the instances in Table 3 where degradation in both utility and safety occurs are from the SeqKD training method, which does not use the SFT distribution. Hence, the difficulty is not due to the SFT distribution being hard to model. The same distribution used by the poorly performing SeqKD is used by KD (FKL), MiniLLM (RKL), and GKD with a different loss function and k-value, but these methods result in higher utility gained at a higher safety cost.
>
> > A fair comparison should have similar utility.
> >
>
> We agree, but similar utility can either be achieved if a) methods result in similar performance for a given model, or b) by creating comparative advantage metrics. As the methods we use are iterative improvements or use fundamentally different elements, a) is not possible. However, we show a fair comparison through Table 2, which represents ratios that show, for every unit of utility gained, how much safety increases or decreases. Similarly, Figure 1 shows, given a baseline safety and utility, the percentage shift observed across methods. Through both these artifacts, we observe soft label distillation yielding higher utility gains at higher safety costs.
>
> > It would be great if the authors can bring some more insights on why they think knowledge distillation could hamper safety performance more than SFT. Currently, it is not clear as in case of using hard labels for knowledge distillation (KD), both SFT and KD would use the same training loss.
> >
>
> While SeqKD and SFT use identical cross-entropy loss, they differ in training data: SFT uses ground-truth labels while SeqKD uses teacher-generated outputs. Our results (Section 5.1, Table 2) show that SeqKD's inferior performance on both dimensions suggests cross-entropy fails to effectively transfer teacher knowledge without soft labels' probability information. For soft-label distillation's amplified safety degradation, we identify three mechanisms: (1) KL-based objectives encode dense utility information from teacher logits while lacking explicit safety constraints, causing models to prioritize task performance over safety behaviors (Section 5.1); (2) soft labels concentrate probability mass on familiar lexical patterns, inadvertently reinforcing the same linguistic structures that safety alignment relied upon for refusal (Section 5.2); and (3) our uncertainty analysis (Section 4.1, Table 4) reveals soft-label methods produce 10× higher epistemic uncertainty on safety evaluations compared to SFT, indicating the rich knowledge transferred through full vocabulary distributions causes models to become uncertain about safety capabilities while maintaining confidence on trained tasks.

---

> ### Author Response · Authors · 2025-12-03
> **Response to Questions**
>
> Thank you very much for the questions provided; we address them below.
>
> > It would be great if the authors can try to use hard label distillation between different families of models. This can help them give effects similar to SFT.
> >
>
> We appreciate this suggestion. While cross-family hard label distillation is feasible (unlike soft labels which require matching tokenizers), our focus is on understanding safety implications of current distillation practices, which predominantly occur within model families. Moreover, our SeqKD results demonstrate that hard label distillation underperforms even within families, suggesting the fundamental limitation lies in discrete labels' inability to capture rich teacher knowledge rather than family-specific effects. We acknowledge cross-family distillation as valuable future work.
>
> > I would recommend that the authors should try to compare their numbers for the same utility performance. Then their results can give a stronger signal. Perhaps reframing their results as demonstrating safety-utility trade-off could be more interesting. This has also been observed recently in a few other works [1]
> >
>
> Our work is explicitly framed as a safety-utility trade-off study from the title onwards. We provide direct comparisons at equivalent baselines through: (1) Table 2's safety-utility ratios showing safety cost per utility unit gained, and (2) Figure 1's percentage changes from shared baselines, enabling fair comparison across methods with different absolute performance. These comparative metrics reveal that soft label distillation consistently yields higher utility at disproportionately higher safety costs (up to 10× worse ratios than SFT), which is the core trade-off we investigate.

---

### Note · Authors · 2026-01-15

**Comment:**

We would like to thank the reviewers and AC for taking time to go through our work. We believe the paper is not a right fit for ICLR, and would like to gracefully withdraw the paper.

**Withdrawal Confirmation:**

I have read and agree with the venue's withdrawal policy on behalf of myself and my co-authors.